# The Influence of Geometry, Surface Texture, and Cooling Method on the Efficiency of Heat Dissipation through the Heat Sink—A Review

**DOI:** 10.3390/ma16155348

**Published:** 2023-07-29

**Authors:** Karol Grochalski, Wojciech Rukat, Bartosz Jakubek, Michał Wieczorowski, Marcin Słowiński, Karolina Sarbinowska, Wiesław Graboń

**Affiliations:** 1Faculty of Mechanical Engineering, Poznan University of Technology, Piotrowo 3, 60-965 Poznan, Poland; wojciech.rukat@put.poznan.pl (W.R.); bartosz.jakubek@put.poznan.pl (B.J.); michal.wieczorowski@put.poznan.pl (M.W.); 2Faculty of Civil Engineering and Transport, Poznan University of Technology, Piotrowo 3, 60-965 Poznan, Poland; marcin.slowinski@put.poznan.pl; 3TRAG Sp. z o.o. Sp.k., Wschodnia 5B, 62-080 Swadzim, Poland; office@trag.pl; 4Faculty of Mechanical Engineering and Aeronautics, Rzeszow University of Technology, 35-959 Rzeszow, Poland

**Keywords:** heat sink, surface asperities, topography, cooling efficiency, heat dissipation, flow simulation

## Abstract

The performance of a heat sink is significantly influenced by the type of cooling used: passive or active (forced), the shape of the heat sink, and the material from which it is made. This paper presents a review of the literature on the influence of geometry and surface parameters on effective heat transfer in heat sinks. The results of simulation studies for three different heat sink fin geometries and cooling types are presented. Furthermore, the influence of the surface texture of the heat sink fins on the heat transfer efficiency was determined. It was shown that the best performance in terms of geometries was that of a wave fin heat sink. When the surface texture was analyzed, it was found that an increase in the amplitude values of the texture decreases the heat dissipation efficiency in the case of active cooling, while for passive cooling, an increase in these parameters has a beneficial effect and increases the effective heat transfer to the surroundings. The cooling method was found to be the most important factor affecting heat dissipation efficiency. Forced airflow results in more efficient heat transfer from the heat sink fins to the surroundings.

## 1. Introduction

The basic design criterion for cooling systems is the maximum operating temperature of the cooled unit. The operating temperature should not be exceeded during typical operation under nominal operating conditions, defined as thermal resistance [1,2,3]. The type of cooling system used—active (forced) [4,5,6,7,8,9,10,11,12,13,14,15,16,17,18,19,20,21,22]/passive [22,23,24,25,26,27,28,29,30,31,32,33,34,35,36,37,38,39,40,41,42,43,44,45,46,47,48,49,50,51] and the selection of material [11,12,13,14,15,35,36,37,38,39,52,53,54] from which the heat exchanger (heat sink) is made—depends primarily on the amount of generated heat, the heating power of the source, and the method of heat transfer to the environment. The ambient conditions in which the cooling system operates are also important: temperature, humidity, flow rate of the cooling medium (e.g., air, water, or other coolants), the possibility of fouling a heat sink with dust and deposits, etc. [13,14,15,16,17,18,19,20,21,22,23,24,25,26,27,28,29,30,31,32,33,34,35,36,37,38,53]. Equally important from this point of view are the texture and topography of the heat sink surface, which are consequences of the treatment applied during manufacturing [4,11,32,33,34,39,40,41,42,43,55,56,57].

Heat sinks have found particular use in electronics [9,10,11,15,21,26,34,35,36,37,47,48,49,50,51,52,58]. However, they are also used in various fields of industry, including the cooling of photovoltaic systems [16], battery thermal management systems [17], platform inertial navigation systems [18] in space engineering applications [19], the cooling of semiconductor chips [22], in air dryers [26], lighting systems [8], and thermoelectric generators [59].

The increasing miniaturization of integrated circuits, combined with their increasing operational capabilities, is resulting in increased heat generation [60,61]. This requires increasingly efficient cooling systems. Originally, electronic circuits were cooled using passive heat exchangers such as bar, plate, and fin heat sinks. The increasing power of microprocessor systems has led to the emergence of forced-flow cooling solutions on the market. An examples of commonly used heatsinks are presented in the Figure 1.

Several papers have been so far published that address the optimization of cooling systems, both experimentally [7,8,9,10,11,12,16,19,26,27,28,29,30,31,32,33,34,38,39,40,41,43,54,56,58,62] and analytically or based on simulations [4,5,6,7,8,9,13,14,15,16,17,18,19,20,22,23,24,25,26,28,29,30,31,32,33,35,36,37,38,39,40,43,44,45,46,47,48,49,50,51,53,54,55,56,57,58,62,63]. Currently, advanced numerical methods based primarily on the finite element method or genetic algorithms are used for this purpose [4,7,9,10,13,14,19,20,29,30,31,32,33,35,36,37,41,42,43,44,45,48,49,50,53,54]. Moreover, several other reviews of heat sinks have also been published so far [59,63,64,65,66].

### 1.1. Heat Sinks in Engineering

One can find quite many papers describing studies of the influence of heat sink geometries on their thermal performance [5,6,7,14,15,16,19,20,21,22,24,25,26,28,29,30,31,32,33,34,35,36,37,38,42,43,44,47,49,58,63].

Kumar et al. performed an analysis of the fluid flow and thermal behavior of air-cooled microchannel heat sinks. Straight, corrugated, and corrugated-branched channel geometries were considered [29]. The thermal-hydraulic behavior was found to be most favorable for a heat sink with wavy-branched channels. Abbas et al. analyzed a heat sink in which alternating fins were extended in opposite directions by a dimension of 20% of the length of the entire heat sink. The experiment showed that the introduced change reduces thermal resistance by 10.4% (ribs facing upwards) and 11% (ribs facing downwards). The authors highlighted the fact that the placement of the ribs significantly affects the thermodynamics of the heat sink. Furthermore, the design changes contributed to a weight reduction of 28.7% [20]. Deng et al. investigated the thermal and hydraulic behavior of double-channel heat sinks dedicated to the cooling of microelectronic devices that emit high heat intensity flux [49]. Heat sinks with triangular, rectangular, trapezoidal, circular, and Ω-shaped cross-sections were considered. The results of the analysis showed that, compared to single-layer microchannel heat sinks (with a triangular cross-section), double-layer ones reduce wall temperature and thermal resistance, resulting in a more homogeneous temperature distribution across the wall, as well as reducing pressure drop and pumping power. The authors noted that the heat sink with a rectangular cross-section had the best thermal properties, while the heat sink with a trapezoidal cross-section had the worst properties in terms of thermal resistance, pressure drop, and pumping power.

Rubio-Jimenez et al. presented a publication in which they investigated the thermal and hydraulic properties of different structural configurations of heat sinks, based on the laws of number Ψ or Allometry [47]. The authors showed that heat sinks with Ψ-shaped channels provide a more homogeneous temperature distribution than Y-shaped ones, while also noting that this comes at the expense of slightly greater flow restriction. Li and Chao considered the effect of the Reynolds criterion number (*R_e_*) of the cooling air, the height dimension, and the width of the heat sink fins on the value of thermal resistance and pressure drops. Based on experimental studies, they found that an increase in the Reynolds number value could cause a decrease in the thermal resistance of the heat sink. Attention was also drawn to a regularity related to the width of the heat sink fins [28]. A change in thickness decreases heat transfer as a consequence of smaller flow channels and smaller heat transfer surfaces. Chiu et al. investigated the heat transfer of a microchannel liquid-cooled heat sink with particular reference to the effect of channel geometry on fluid flow [30]. Bornhoff and Parry addressed the fabrication of heat sinks using 3D printing technology, which enables more sophisticated geometries than conventional methods such as casting. The authors suggest that new and more complex heat sink geometries may improve the thermal properties of heat sinks [42].

When considering the issue of heat sink optimization in terms of thermal performance and weight reduction, the following work should be highlighted. Wu et al., presented an automated method for the optimization of heat sink geometries, based on a genetic algorithm and finite element analysis. They were guided by the criterion of the lowest temperature obtained at the interface. Based on a comparative analysis with a conventional heatsink design, the authors found that the optimized heatsink geometry provided up to a 6% reduction in junction temperature with a 27% reduction in size [40]. Ning et al. optimized mathematically the heat sink geometry to reduce its weight [31]. Pakrouh et al. described the optimization of the geometry of a heat sink whose fins were made of aluminum and a phase change material. The main objective of the study was to find a configuration that ensured the longest possible operating time of the heat sink. An analysis of different heatsink configurations was performed using both statistical and numerical methods [5]. Husain and Kim presented a multi-criteria optimization of the shape of a microchannel heat sink using numerical analysis. In this case, the objective function was described by thermal resistance and pumping power [67].

Interesting scientific contributions in the context of radiation-based heat transfer methods are provided by publications describing idea fronts aimed at improving the efficiency of heat sinks used in cooling systems for computer processors or integrated circuits. Annuar and Ismail presented an analysis of different fin arrangements of a heat sink used to cool a processor, in terms of its thermal performance [50]. They proposed a new fin arrangement that allowed a temperature reduction of 2.84% (inline arrangement) and 0.63% (offset arrangement) compared to the conventional design. The research team, led by Choi, proposed a new processor cooling system design using active cooling via a heat sink. Based on their research, the authors showed that the changes made reduced the thermal resistance values while reducing the sound level [48]. Moradikazerouni et al. presented a thermal analytical study under forced laminar convection conditions of heat sinks used for processor cooling. Based on the results, it was indicated that longer fins and more fins have a significant effect on the heat transfer between their surface and the coolant [45]. In turn, Brench, in a publication [21], discussed the effect of the geometry of heat sinks that are used to cool processors or integrated circuits on their heat radiation.

Another group of papers is a series of publications treating materials used in the construction of heat sinks. Ekpu et al. presented the choice of construction material for heat sinks used on laptops. Currently, the most commonly used materials are aluminum (low weight criterion) and copper, which are used where weight reduction is not required [52]. In this study, an analysis was conducted that suggested a composite consisting of aluminum and silicon carbide (Al/SiC) as a new material for the construction of heat sinks used in laptops. The Yamada et al. team investigated the thermal properties of heat sinks consisting of graphite plates coated with a thin layer of copper. The team found that the new laminated heat sink design had a lower thermal resistance than those made of copper. In addition, it was pointed out that due to its properties, this design has potential for application in semiconductor devices [54]. Oguntala et al. analyzed porous heat sink fins made of gradient plastics and used them to improve thermal properties [56].

There are also publications in the literature directed at studying the properties of heat sinks. Basyigit et al. presented an algorithm using the Deep Learning technique to predict resonant frequencies in heat sinks with a two-parallel plate–rib arrangement [58]. Mokheimer presented an analysis of the effect of a locally varying heat conduction coefficient on the performance of heat sink fins under free convection conditions. A study of heat sinks with different geometries was carried out [6]. He and Hubing determined a closed-form expression to calculate the maximum value of radiation from a Printed Circuit Board (PCB) or base plate heat sink as a function of the maximum voltage between the heat sink and the surface [15]. Shen et al. describe a measurement of maximum radiated emission for a heat sink-IC system [9]. Liu et al. investigated the properties of innovative heat sinks with double-layer die structures [24].

An interesting approach to the evaluation of heat sink performance was presented by Li, Wang, and Deng. The Reconstructive Neural Network model has been suggested for the reconstruction of heat transfer processes to construct more objective and more information-included models. The research team states that these models can be further applied to research or some experiments that use high-speed photography to catch some dynamic characteristics and reconstruct the model through these photos [59].

A separate group of publications on heat sinks can be formed from papers describing the use of nanofluids as a heat sink coolant. Shahsavar described the flow of a nanofluid mixture consisting of biological water–Ag nanofluid through helical microchannels of a heat sink [25]. Ma et al. also conducted a study of the flow of this type of nanofluid in a heat sink with an emphasis on evaluating the effect of the heat sink fin arrangement on hydrothermal behavior [36]. Rajabifar conducted a study to analyze the performance of a double-layer microchannel heat sink when its cooling is carried out through nanofluid and Phasce Change Material (PCM) slurry coolants [46].

In reviewing the work on heat sinks, mention should also be made of the work describing heat sinks operating using phase transformation of materials. A description of heat sinks using phase transformation of metallic foams can be found in the following works. Lee et al. investigated a flexible heat sink using phase transformation, which could find application in mobile thermoelectric generators for wearable devices [12]. Ho et al. presented a description of convective heat transfer in a microchannel heat sink using Nano-Encapsulated Phase Change Materials (NEPCM) [27,38]. Dammak and El Hami presented the results of the transient 3D numerical simulations of a pin fin heat sink filled with phase change material (paraffin wax). They have studied the heat transfer performance for passive cooling of electronic devices [51].

It should also be noted that there are texts that describe the use of heat sinks as antennas. Casanova et al. carried out simulations directed at studying the thermal and electromagnetic properties of a heat sink that is also an antenna [63]. Covert et al. presented the use of a heat sink as an antenna, with an emphasis on analyzing its performance depending on the geometrical arrangement of its fins [10].

### 1.2. Surface Asperities and Heat Sink Performance

In the literature, some papers focus on analyzing the influence of surface roughness on the heat performance of heat sinks. Undoubtedly, they constitute an interesting research contribution; therefore, the most important achievements on this issue will be presented in this section.

Dai et al. [55] are the authors of the analysis which was focused on the effect of roughness on a flow in micro-and mini-scale channels. Existing prediction models were used to predict the friction factors of fully developed flow and the critical Reynolds number—*R_e_*. The most important conclusion of this work is that roughness had little effect on flow characteristics when the relative roughness is less than 1%, but for the relative roughness greater than 1%, the friction factor and the critical Reynolds number are divergent from the prediction values of smooth tubes.

Alam et al. [34] experimentally investigated the influences of surface roughness on flow boiling heat transfer, pressure drop, and instability in the microgap heat sink. Three microgap dimensions were selected to conduct flow boiling experiments: 500 µm, 300 µm, and 200 µm. To examine the effect of surface finish on thermal performance, the original surface roughness of the silicon, *Ra* = 0.6 µm was modified to *Ra* = 1.0 µm and 1.6 µm. Deionized water at the temperature of 91 °C and two different mass fluxes were chosen as the medium in an experiment. The study provided the following research conclusions:The density of the bubble nucleation site increases with increasing surface roughness from *Ra* = 0.6 µm to 1.0 µm for a larger microgap heat sink;*Ra* = 1.0 µm and 1.6 µm implies a similar boiling behavior of microgaps;The increment of the surface roughness implies the increment of the local heat transfer coefficients;During the experiment, there was no observed significant adverse effect of surface roughness on the pressure drop curve;*Ra* = 1.0 µm and above provides a more uniform wall temperature over the heat sink compared to *Ra* = 0.6 µm, due to the uniform distribution of the boiling process over the heating surface;*Ra* = 1.0 µm and 1.6 µm do appear to harm the inlet pressure instability compared to *Ra* = 0.6 µm at a larger microgap heat sink, and *Ra* = 1.0 µm and 1.6 µm surfaces have between 20% and 40% higher inlet pressure instability than *Ra* = 0.6 µm surface;Wall temperature fluctuations are independent of surface roughness for the microgap at various imposed heat flux and mass flux.

Lu et al. [4] made a numerical analysis of the effects of surface roughness on laminar flow and heat transfer in microchannels. Three types of channels were selected for this study: square channel, wavy channel, and dimpled channel. Surface roughness was modeled as a superposition of waves, which were described by the Fourier series and exponential correlation function. A relative roughness of up to 2% was introduced to one of the four side walls. The surface roughness was introduced by meshing a smooth channel. The constant value of the Reynold number (*R_e_* = 500) and the heat flux (Irriadance surface *E_e_* = 0.5 W/mm^2^) were applied to the bottom of the heat sink. The results of the analysis showed that roughness could increase the overall flow resistance and the Nusselt number—Nu*_L_*. Moreover, the authors highlighted the fact that the local influence of roughness strongly depends on the geometric shape of the microchannel.

Sterr et al. [33] numerically investigated the stochastic heat transfer performance of a microchannel heat sink. A randomly generated rough surface profile with a prespecified autocorrelation function was applied to the bottom surface of the heat sink. The magnitude of the maximum relative roughness of the surface was treated as a Gaussian random input with specified uncertainty. Uncertainty in the output was modeled via polynomial chaos expansions. The study showed that an increment in surface roughness enhanced the heat transfer. According to the study, the performance factor of the heat sink increased monotonically with increasing relative roughness. The authors suggested that the positive effect of the surface peaks on the heat transfer outweighs the negative effect of the valleys, as an explanation for the aforesaid phenomenon.

Ventola et al. [39] compared the thermal performance of copper heat sinks produced by traditional milling with aluminum alloy heat sinks (AlSi10Mg) produced by Direct Metal Laser Sintering (DMLS). The authors emphasized that *DMLS* technology enables the production of heat sinks whose surface roughness can vary over a wide range of values. The result showed that *DLMS* technology implied 50% and 20% cooling enhancement in the flat and finned heat sink, respectively, in comparison to conventionally milling samples.

Oguntala et al. [56] analyzed the Chebychev spectral collocation method a numerical study focused on the thermal behavior and subsequent heat transfer enhancement of cylindrical micro-fin heat sink with artificial surface roughness. The study was conducted to establish the thermal performance of the rough fin over the existing smooth fin. In general, the authors outlined that the surface roughness of the fin enhances its thermal performance.

You et al. [41] are the authors of the research on the effect of surface microstructure on the thermal performance of heat sinks used in electronic devices. Four types of treatment methods (polishing, chemical coarsening, mechanical shot peening, and chemical oxidation) were applied to make different microstructures on heat sink surfaces. The study shows that the increase in the surface roughness caused up to 2.5 times increase in Thermal Emissivity—*ε*.

Hoang et al. [32] presented an experimental study on the two-phase cooling heat sink. One of the objectives was to investigate the enhanced surface roughness for thermal performance. The authors indicated that untreated surfaces imply ineffective heat transfer. For this reason, the original copper surface was modified to achieve higher roughness by using sandpaper. It has been found that increasing the roughness of the bare copper surface can lead to an increase in the heat transfer coefficient *α* of 9% compared to the bare copper.

Guo et al. [43] compared the novel Gauss rough surface model with 3D and 2D models from the literature. One of the authors’ conclusions is that flow resistance and heat transfer are sensitive to the surface morphology of microchannels. Moreover, they indicated that roughness plays a positive role in thermal performance and flow resistance in the case of laminar flow.

In the paper [57], Koo and Kleinstreuer selected the relative surface roughness (SR), represented by a Porous Medium Layer (PML) model, as, a key microscale parameter in heat analysis. The study showed that the effect of surface roughness on heat transfer is less significant than on momentum transfer.

This paper aims to review and independently compare the performance of the cooling process for three different heat sink fin designs, when passive and active cooling is used, and to verify the impact of the different surface texture quality, important from a manufacturing and economic process point of view. The paper presents the results of simulation studies of the cooling process for different surfaces. Two numerical experiments were carried out for each of the heat sink variants analyzed. The first concerns the case of passive cooling, while the second was related to the active cooling process through an air stream flowing parallel to the base of the heat sink along the side surfaces of the fins.

## 2. Methodology

Due to the nature of the environment in which the numerical experiment was conducted, all three types of thermal energy transfer must be considered:Conduction—at the interface between the source and the heat sink and in the volume of the heat sink itself;Radiation—on the surface of the heat sink;Convection—in the volume of the environment surrounding the heat sink.

The mathematical model describing the phenomena occurring in the analyzed problem depends on the considered point in space—the heat sink (its volume/surface area), the environment, which determines the type of heat transfer occurring at this point.

The steady-state heat conduction equation in a continuous medium is described by the Fourier–Biot equation. The form of this equation is given by the following Equation (1).
(1)$$\frac{\partial }{\partial x}\left(k_{x}\left(T\right)\frac{\partial T}{\partial x}\right)+\frac{\partial }{\partial y}\left(k_{y}\left(T\right)\frac{\partial T}{\partial y}\right)+\frac{\partial }{\partial z}\left(k_{z}\left(T\right)\frac{\partial T}{\partial z}\right)+Q=0$$
where: *k_x_*(*T*), *k_y_*(*T*), *k_z_*(*T*)—directional heat conduction coefficients [W/m·K], if the material is isotropic, then *k_x_*(*T*) = *k_y_*(*T*) = *k_z_*(*T*) = *k*(*T*); *Q*—heat generation rate [W/m^3^]; *T*—temperature [K].

The energy balance equation—the flow of thermal energy, by way of radiation is described by the following relationship (2).
(2)$$R_{n}=S_{t}+S_{e}-\rho_{b}\left(S_{t}+S_{e}\right)+\varepsilon \left(L_{d}+L_{e}\right)-\varepsilon L_{b}$$
where: *R_n_*—total radiation flux; *S_t_*—component of direct and diffuse solar radiation flux in the atmosphere; *S_e_*—component of solar radiation flux reflected by the surroundings; $\rho_{b}$—surface reflectance; ε—emissivity of the surface; *L_d_*—radiation flux of the atmosphere; *L_e_*—radiation flux of the surroundings; *L_b_*—radiation flux of the blackbody.

The energy balance for convective energy transfer is given by Equation (3).
(3)$$q_{s}=\alpha \left(T_{s}-T_{p}\right)$$
where: *q_s_*—heat flux density [W/m^2^]; *α*—heat transfer coefficient [W/mK]; *T_s_*—body surface temperature [K]; *T_p_*—fluid temperature [K].

It is known that the roughness of the surface affects the local nature of fluid flow near the wall of the structural element—the cooling fins of the heat sink—and, thus, plays a role in the process of intensification of the flow (reception) of heat to the environment.

This was confirmed in the work of Kinal and Gęstwa [68]. The authors showed that increasing the surface roughness of the workpieces before quenching by sandblasting them beforehand favorably influences the rate and intensity of cooling during quenching, which in turn translates into a higher hardness of the surface after heat treatment.

The effect of surface roughness on the conditions, such as rate, intensity of heat transfer, and other aspects of thermal energy transfer, has been addressed many times to date by various research teams. In his work, Maisuria showed that as the roughness of the surface plating increases in the case of physical contact between them, the rate of heat transfer decreases (reducing the actual area of direct contact between the bodies) and the loss of heat to the environment increases [69].

In turn, a team led by Azri bin Alias showed that as roughness increases—in the range of 0.1 to 10 µm—and the thermal conductivity of the contact bodies increases, the rate of heat transfer increases, and the time required to reach local thermal equilibrium at the point of contact decreases [70].

Subramanian et al. showed that as the roughness of the surface increases, the local flow resistance of the cooling medium increases. Stabilized laminar flow under extreme conditions can turn into turbulent flow, dramatically worsening cooling conditions [71]. However, Bai T. et al. showed that as the roughness of the surface increases, the flow resistance increases—this is related to the local increase in Reynolds number *R_e_*. These losses negatively affect the intensity of active cooling [72].

In this paper, simulation studies of 18 variants of heat sink models differing in geometry and surface texture were performed for each of two different types of cooling—passive and active. The digital model of the experimental space and simulations were created in the Solidworks Flow Simulation engineering environment, while the discretization of the computational model was performed according to the principles and assumptions of the Finite Element Method (FEM).

The heat transfer was modeled using the Surface-to-Surface model (S2S). In a S2S heat transfer model, all the solid surfaces are assumed to be gray and diffuse, meaning that their emissivity and absorptivity are independent of the wavelength and the reflectivity is independent of the outgoing or incoming directions. This means that any absorption, emission, or scattering of radiation can be ignored; therefore, only “surface-to-surface’’ radiation needs to be considered for analysis [73,74,75].

The calculation model included the following:The heat source—a processor system of known power and dimensions;The heat removal element—a heat sink;Workspace or flow channel.

The evaluation and comparison of the models considered were made on the basis of the following criteria:Temperature distribution on the surface of the heat sinks;Distribution of velocity vector fields and airflow directionality in the heatsink surroundings;Effective heat dissipation efficiency.

The heat source is a typical, commonly-used processor chip, whose operating temperature was set at 80˚C. Other parameters of the source are given in Table 1 [76,77].

The working space of the heat sink for passive cooling is an open space, while for active cooling, it is a tubular duct. For this case, a free flow of the cooling medium—air—is provided. Parameters related to forced air flow (e.g., through a fan) are given in Table 2 [76,77].

The structures tested were characterized by different ribbing methods and different geometric surface structures. The first way of ribbing is straight ribs of equal height; the second way is straight ribs of different heights arranged alternately. The third way is a wave-shaped rib with equal height ribs. The values of the individual dimensions marked in Figure 2 for the heat sink models in comparison are summarized in Table 3.

The heat sinks models made, based on the dimensions summarized in Table 3, are shown in Figure 2.

To compare the analyzed heat sink models with each other, the same boundary dimensions and the same type of material were used for each heat sink. The heat sinks bases had the same height and surface area. The heat sink material was defined as aluminum 6061 (AlMg1SiCu) with known physical parameters. A summary of the designations of the heat sink variant is shown in Table 4.

Each of the ribbing methods was also characterized by six types of surface structure. The surface texture was modeled as it corresponds to the milled surface. The ground surface (*xR*) was taken as the reference texture of the heat sink fin. It should be noted that milling is not used in the manufacture of heat sinks. Such surfaces were used in the modeling due to the convenience of the detailed analysis of their structure performed previously.

The *FEM* mesh of the geometric structure was generated using tetra- and hexahedral volume elements (tetrahedrons and hexahedrons). To ensure high convergence (overlap) of the results obtained by simulation, isoparametric adapted elements of complex type were selected: 10-node and 20-node elements (for 4- and 6-wall elements, respectively). As a result, consistency was preserved: constancy and changes in the magnitude of thermal parameters considered in the framework of a single finite element (nodal points) as well as the entire continuum—temperature distribution and gradient. Interpolation polynomials to calculate the values of the above parameters for the entire continuum are described by the following relations. Formula (4) is used for 4-wall elements, while (5) is used for 6-wall elements.
(4)$$T^{e}{}_{\left(x;y;z\right)}=\alpha_{1}{}^{e}+\alpha_{2}{}^{e}x+\alpha_{3}{}^{e}y+\alpha_{4}{}^{e}z+\alpha_{5}{}^{e}xy+\alpha_{6}{}^{e}yz+\alpha_{7}{}^{e}xz+\alpha_{8}{}^{e}xyz+\alpha_{9}{}^{e}x^{2}+\alpha_{10}{}^{e}y^{2}+\alpha_{11}{}^{e}z^{2}$$
(5)T(x;y;z)e=α1e+α2ex+α3ey+α4ez+α5exy+α6eyz+α7exz+α8exyz+α9ex2+α10ey2+α11ez2  +α12ex2y+α13ey2z+α14ez2x+α15ex2yz+α16ey2xz+α17ez2xy+α18ex2y2z  +α19ey2z2x+α20ex2z2y+α21ex2y2z2
where: *e—*the unit version of the local coordinate system

A view of the surface structures modeled on the heat sink surface is shown in Figure 3.

Selected spatial surface texture parameters (mean values) of the heat sink fins are presented in Table 5.

The surface structure parameters presented and modeled correspond to real milled surfaces made with a tool with a corner rounding radius *r* = 0.4 mm, tool speed *v_c_* = 1000 rpm, and feed rate *v_f_* in the range of 100–500 mm/min. The reference surface was given surface parameters corresponding to typical abrasive machining performed at a grinding wheel speed of 1000 rpm for a grinding wheel with an average grain size of 400.

## 3. Results

This chapter presents the results of the numerical experiments carried out. First, the results related to the passive cooling case for all geometries are presented, for the reference surface texture. Then, the results related to the active cooling case are presented. The influence of surface texture on the temperature value occurring on the surface of the heat sink is presented in the graphs.

The distribution of the temperature values on the surface of the heat sinks tested depending on the type of fining for the passive cooling process is presented in Figure 4.

The distribution of temperature values for passive cooling is symmetric to the heat source. The highest temperatures occur at the base in the central part and decrease with distance from the heat source. Figure 5 summarizes the maximum temperature values occurring on the heat sink depending on the surface texture.

In the case of passive cooling, depending on the chosen heatsink geometry, the maximum temperature recorded on the surface of the fins reaches between 79.81 °C and 76.43 °C. In all cases, the lowest recorded temperature is for the wave fin heatsink, which is due to the largest radiation surface area. It is worth noting that, depending on the surface structure, the effective heat exchange efficiency of the heat sink with the surroundings also changes. The higher the value of the surface roughness, the more effective the convective heat transfer in the case of passive cooling. It is worth noting, however, that these differences are relatively small, up to 3 °C.

The directions of the vector field of the air stream flowing around the tested heat sinks, together with a color representation of the temperature value of the air stream, in the case of passive cooling, are presented in Figure 6.

As can be seen from Figure 6, for the passive cooling mode, in each case, the highest airflow temperatures occur around the top of each heat sink. The difference in air temperature at different points in the space between the heatsink fins, concerning both the longitudinal dimension of the inter-rib gap and the height dimension, results in the so-called chimney effect [7]. The consequence of this is the circulation of cooler air masses located at the base of the heat sink to the intercostal spaces and upward. The graphical representation of the flow simulation for the variants analyzed with different surface structures did not show noticeable differences in the flow of the cooling medium.

In the second variant of the study, simulations related to the active cooling process were carried out for the heatsink variants considered, taking into account their geometry and surface texture.

The distribution of temperature values on the heat sink surfaces for active cooling is presented in Figure 7.

The distribution of temperature values for active cooling is asymmetric concerning the direction of airflow. The highest temperatures occur (as in the case of passive cooling) at the base of the heatsink; however, a higher value was recorded at the rear, directly unaffected by the flowing air. The lowest temperature values were recorded at the front of the heatsink, particularly at the edges and tops of the fins.

Depending on the surface texture, the maximum temperature values occurring in the heat sink are summarized in Figure 8.

Particularly noteworthy is the fact that, depending on the surface structure, the effective heat exchange efficiency of the heat sink with the surroundings changes to a noticeably higher degree than with passive-convection cooling. The higher the value of the surface roughness, the lower the heat exchange efficiency in the case of active cooling. This effect is caused by the resulting turbulence or local pressure changes on the surface of the heat sink fins, which are associated with a reduction in air velocity and higher temperatures on the heat exchanger. These differences reach up to 6.8 °C. An example of turbulence and air velocity distribution on the heat sink fin surface (single element view) is shown in Figure 9.

As can be seen in the figure, the flow velocity of the medium on the heat sink fins decreases. This results in less efficient heat transfer. Furthermore, the additionally occurring non-laminar flow impedes the heat transfer from the fins to the air. The higher the value of the surface roughness, the more noticeable this effect is, which confirms the correlations shown in works [4,32], however, relating to microchannel heat exchangers and the effect of Reynolds number on heat transfer.

In contrast to passive cooling, the main difference is the nature of the temperature distribution on the surface of the heat sinks. The highest temperature values are observed at the rear, directly covered part of the base of the heat sinks—on the side opposite the direction of the incoming cooling medium. The lowest temperature values were recorded for the heat sink with straight ribs of equal height and with corrugated ribs. The areas of the heat sink with the lowest recorded surface temperature are located in the upper part of the fins, mainly on the side of the direction of inflow of the coolant flow and on the edges.

For the cases in question, the minimum temperature is 67.6 °C and the maximum temperature does not exceed 74 °C for the heat sink with wavy fins. Heat sinks with straight fins had the least favorable heat dissipation, with a minimum temperature of 72.3 ° C and a maximum temperature of 79.4 °C.

The nature of the temperature distribution depends on the location of the point under consideration—whether it is on the surface of a fin located on the inside of the heat sink geometry, or whether it is a fin that directly forms the side surface of the entire heat sink. This is particularly evident in the attitude of the heat sinks. As one moves away from the front surface, relative to which the direction of the cooling medium flow is considered, a non-linear temperature increase is observable on the side surfaces—mainly in the lower part of the heat sink. It should also be noted that, with active cooling, it is the flow of the medium itself that is of greater importance and not the geometry of the fins. The geometry, however, essentially determines the air resistance and the pressure difference upstream and downstream of the heat exchanger, directing the flowing media streams accordingly. It should be noted that the value of the heat transfer coefficient *α* depends on the position of the point at which it is determined. The highest value was recorded for a heat sink with corrugated profile fins—on the front surfaces of the individual fins, which are located on the side from which the cooling medium flow pressure occurs. For this type of design, the value of heat transfer coefficient *α* is 85 W/m^2^·K. Heat sinks with straight fins are characterized by lower heat transfer coefficient values. An exchanger equipped with ribs of equal height provides a maximum coefficient value α of approximately 75.6 W/m^2^·K. On the contrary, for a heat sink equipped with straight fins of alternating height, the coefficient α is a maximum of 66.1 W/m^2^·K.

These values considered after the thermal stabilization state has been reached [78,79] allow us to confirm that the heat sink with corrugated fins has the highest dissipation efficiency. The design with straight ribs and alternating rib heights proved to be the worst in this respect. This is related to the local turbulence that causes a local increase in flow resistance.

## 4. Conclusions

In the case of active cooling, the maximum temperature reached by the heat sinks is in the region of 80 °C, recorded close to the base and bottom of the structure. In contrast to passive cooling, for active cooling, there are clear differences in the distribution of temperature values on the heat sink surfaces. The lowest temperature was observed on the surface of heat sinks with corrugated and straight fins of different heights, with a minimum temperature of 66.7 °C.

For the heatsink fin geometries analyzed, the efficiency of the cooling process is higher for each design when active cooling is used compared to passive cooling.

The results of simulation studies clearly show that, in terms of cooling performance, the wave fin heat sink has the best performance. The highest values of the heat transfer coefficient *α* were recorded for this type of design. They amount to 19.4 W/m^2^·K for passive cooling and 85 W/m^2^·K for active cooling, respectively.

The heat sink with straight fins of alternating height has the lowest heat dissipation efficiency. For this design, the heat transfer coefficient *α* is 15.2 W/m^2^·K for passive cooling and 66.1 W/m^2^·K for active cooling.

The distribution of the heat transfer coefficient values for active cooling, in contrast to the passive cooling process, is characterized by the lack of symmetry of the distribution. This relationship is also true for different values of the amplitude parameters of the surface texture and, as the amplitude increases, the heat transfer efficiency decreases.

If the surface texture amplitude values increase, the effective heat removal efficiency for active cooling decreases. This is related to the turbulent airflow in the close vicinity of the heat sink fins, local pressure drops conditioning a reduced cooling media flow velocity and thus a lower heat transfer efficiency.

It is noted that the dependence of the effective heat transfer as a function of the amplitude parameters of the surface texture is non-linear logarithmic for active cooling and non-linear exponential for passive cooling.

For passive cooling, an increase in the amplitude parameters of the surface texture has a positive effect on the effective convective heat transfer to the surroundings. This is due to the larger active radiative surface area of the heat sink. However, in this case, the differences between temperature and surface roughness are not as significant as in the case of active cooling.

The most important factor that influences the efficiency of heat dissipation is the type of cooling used. Forced airflow results in more efficient heat exchange from the heat sink fins.

Verification of the correctness of the results obtained through digital calculation methods is a complex issue. It is necessary to bear in mind the error of the mathematical model *εs* for which an exact solution is obtained concerning the result that would be obtained if the measurements were carried out physically on the real object. In addition to the mathematical error—related to the mathematic apparatus—the error of the discrete model *εd* related to the number of degrees of freedom of the digital model has a significant impact on the accuracy of the calculation.

The number of degrees of freedom determines the number of finite elements and nodes for which the values of the desired parameters are calculated.

Verification of the assumptions of the FEM model and its correctness was carried out by gradually compacting the partition grid for successively implemented simulations.

## Figures and Tables

**Figure 1 materials-16-05348-f001:**
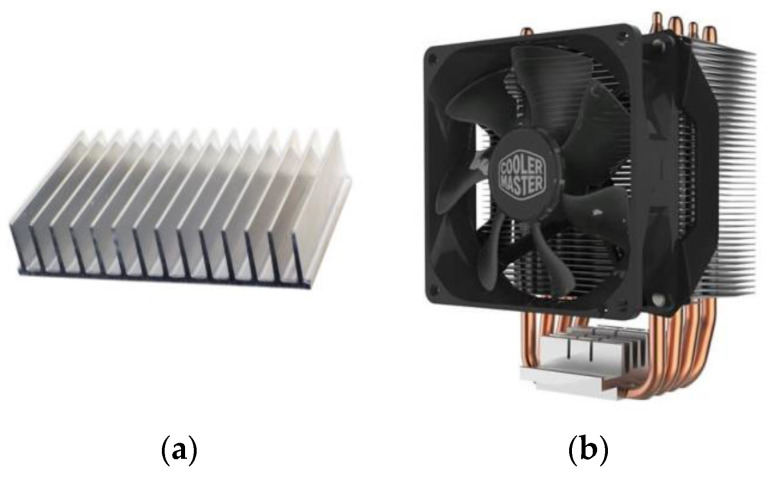
Types of commonly used cooling methods: (**a**) passive heat sink, (**b**) heat sink with active cooling.

**Figure 2 materials-16-05348-f002:**
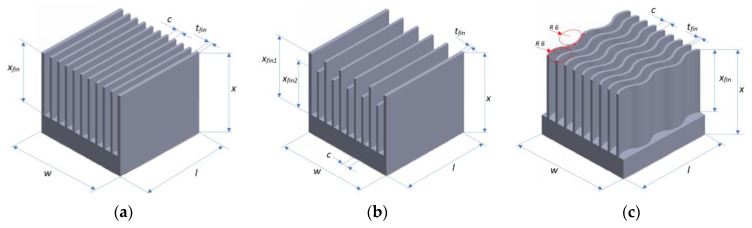
The geometry of the heat sinks considered: (**a**) heat sink with straight ribs of equal height, (**b**) with straight ribs of two heights, and (**c**) with wave-shaped ribs of equal height.

**Figure 3 materials-16-05348-f003:**
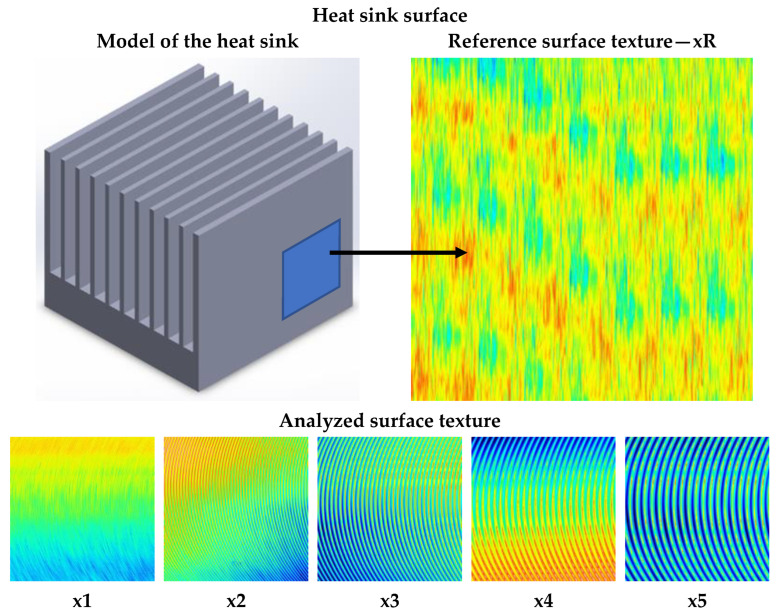
Presentation of the modeled surface structure on the fins of a heat sink with known parameters.

**Figure 4 materials-16-05348-f004:**
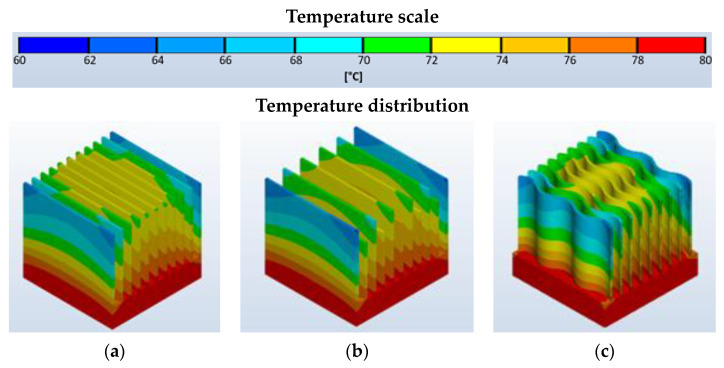
Distribution of temperature values on the surface of the heat sinks for the passive cooling process for the heat sink with straight ribs of equal height—(**a**), with straight ribs of two heights—(**b**), and the heat sink with wave-shaped ribs of equal height—(**c**).

**Figure 5 materials-16-05348-f005:**
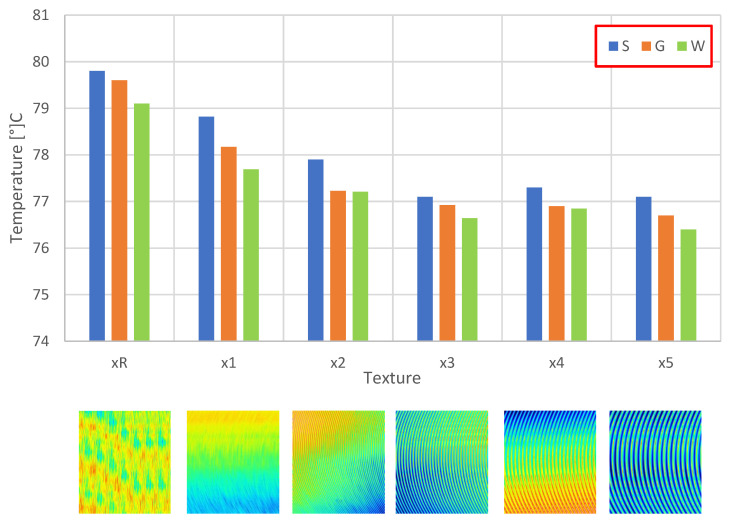
Distribution of heat sink surface temperature values for the passive cooling process as a function of surface texture.

**Figure 6 materials-16-05348-f006:**
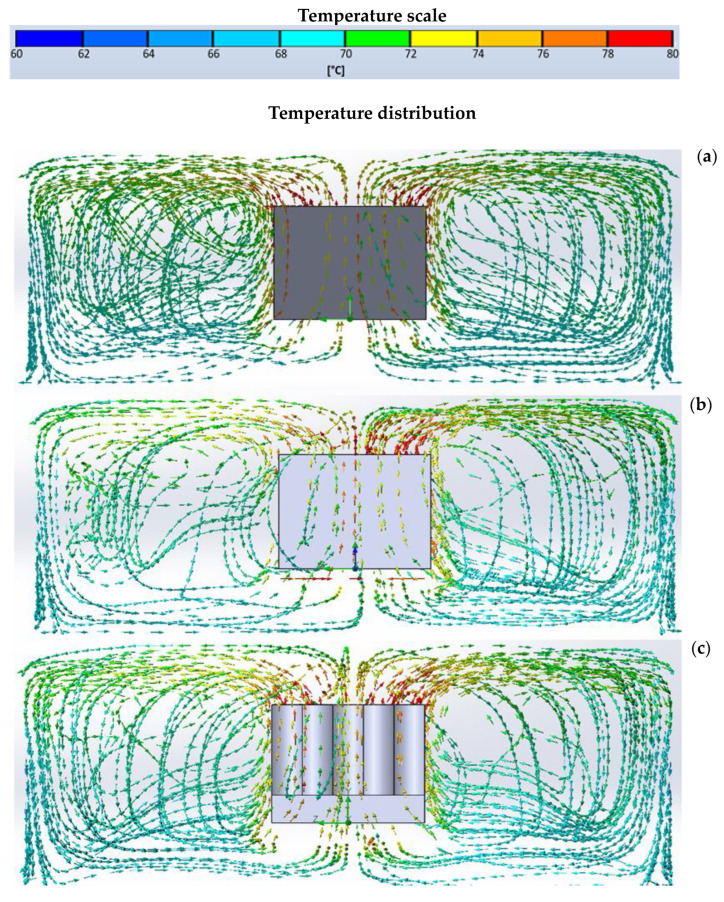
Directionality of airflows (convective flow) for passive cooling for heat sink with straight ribs of equal height—(**a**), with straight ribs of two heights—(**b**), and heat sink with wave-shaped ribs of equal height—(**c**).

**Figure 7 materials-16-05348-f007:**
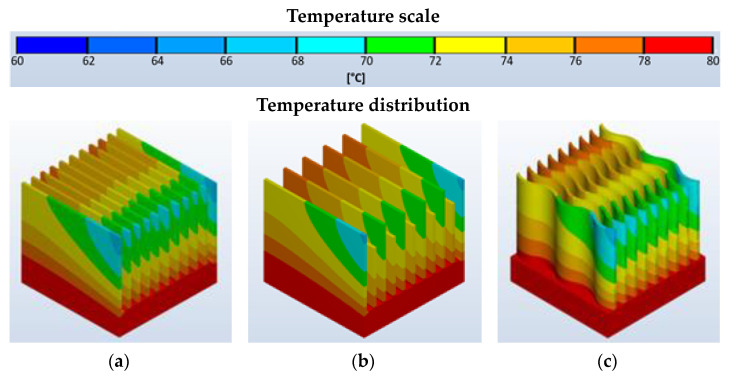
Distribution of temperature values on the surface of the heat sinks for the active cooling process for the heat sink with straight ribs of equal height—(**a**), with straight ribs of two heights—(**b**), and the heat sink with wave-shaped ribs of equal height—(**c**).

**Figure 8 materials-16-05348-f008:**
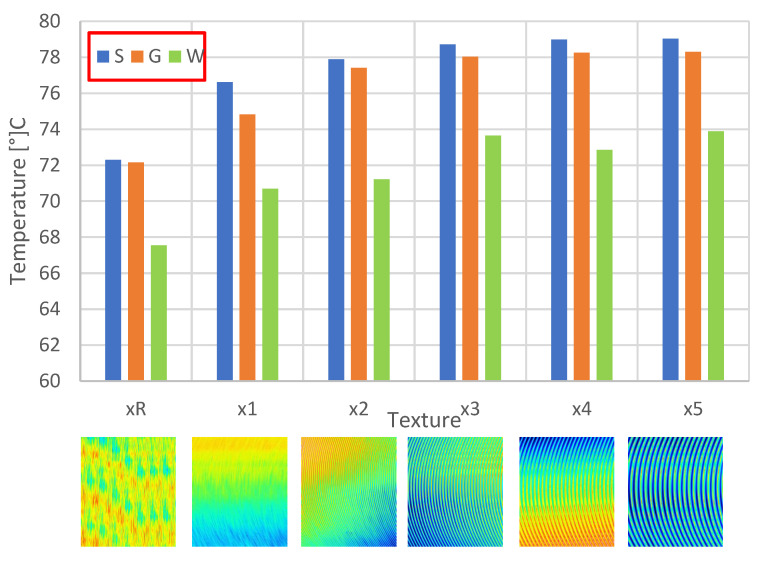
Distribution of heat sink surface temperature values for the active cooling process as a function of surface texture.

**Figure 9 materials-16-05348-f009:**
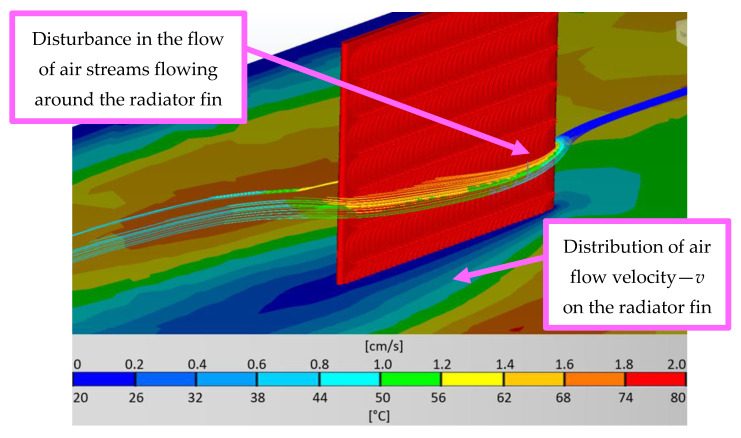
View of the turbulence and heat absorption of the air surrounding the fin (selected) heat sink, together with the plane representing the velocity of the flowing medium.

**Table 1 materials-16-05348-t001:** The technical parameters of the heat source adopted for the simulation.

Material Density —*ρ* [kg/m^3^]	Specific Heat Capacity—*c* [J/(kg*K*)*]	Conductivity Type	Thermal Conductivity —*k* [W/(m*K*)*]	Emissivity —*ε*
2707	896	Isotropic	2.04	0.2

**Table 2 materials-16-05348-t002:** Boundary conditions of the simulation.

	Passive Cooling	Active Cooling
Atmospheric pressure—*p* [Pa]	101,325	101,325
Air temperature—*t* [°C]	20.00	20.00
Source temperature—*t* [°C]	80.00	80.00
Cooling fan revolution speed—ω [rad/s]	-	628
Volumetric air flow rate—*Q* [l/min]	-	90
Maximum airflow velocity—*v* [m/s]	-	0.02
Mesh size—[mm]	0.18	0.18

**Table 3 materials-16-05348-t003:** Geometrical dimensions of the heat sink structures analyzed for the numerical experiment.

Heat Sink with Straight Fins of Equal Height						
*x* [mm]	*w* [mm]	*l* [mm]	*c* [mm]	*t_fin_* [mm]	*x_fin_* [mm]	
34	37.5	37.5	2.6	1	26	
**Heat sink with straight ribs with two alternating heights**						
***x* [mm]**	***w* [mm]**	***l* [mm]**	***c* [mm]**	***t_fin_* [mm]**	***x_fin 1_* [mm]**	***x_fin 2_* [mm]**
34	37.5	37.5	2.6	1	26	20
**Heat sink with wave-shaped fins of equal height**						
***x* [mm]**	***w* [mm]**	***l* [mm]**	***c* [mm]**	***t_fin_* [mm]**	***x_fin_* [mm]**	***r* [mm]**
34	37.5	37.5	3	1	26	6.5

**Table 4 materials-16-05348-t004:** Designations of heat sink variants.

		Surface Texture					
		Reference—Polished *xR*	*x*1	x2	*x*3	*x*4	*x*5
Geometry	Straight (*S*)	*SR*	*S1*	*S2*	*S3*	*S4*	*S5*
	Graduated (*G*)	*GR*	*G1*	*G2*	*G3*	*G4*	*G5*
	Wave (*W*)	*WR*	*W1*	*W2*	*W3*	*W4*	*W5*

**Table 5 materials-16-05348-t005:** Surface parameters of the heat sink models used.

	*Sq*	*Sp*	*Sv*	*Sz*	*Sa*
	[µm]	[µm]	[µm]	[µm]	[µm]
** *xR* **	0.87	4.03	4.47	8.50	0.69
**x1**	3.18	13.78	10.12	23.90	2.69
**x2**	6.08	19.00	16.37	35.36	4.98
**x3**	9.35	36.62	19.14	55.76	7.66
**x4**	13.21	65.00	22.62	87.62	10.51
**x5**	23.88	66.40	57.23	123.63	19.91

## Data Availability

Data is contained within the article.

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
