# Peer review of "The Influence of Geometry, Surface Texture, and Cooling Method on the Efficiency of Heat Dissipation through the Heat Sink—A Review"

_materials, 2023, doi:10.3390/ma16155348_

Round 1

Reviewer 1 Report

1. I am bit confused about the title of manuscript. Is it review article or research article? Authors presented some research results in manuscript after giving some literature review. Please have a better title.

2. Table 1 and Table 2: provide the references for properties.

3. Fig.1(a) and Fig.2 are kind of same. Please have only one figure.

4. Table3: Any criteria to define geometrical dimensions of the heat sink? How did author define them please provide explanations.

5. In Fig. 3, show all the defined parameters in Table3.

6. Some Nomenclature are not provided in manuscript. It is difficult to know the meaning of parameters author used. Could author provide a Table for  Nomenclatures?

8. Line 423: The graph - Figure 6. It should be only Figure (6)

7. what is dotted line in Fig.6 and Fig. 10.

9. Author considered all the mode of heat transfer. Could you provide the contribution of each mode for heat transfer? It is important to support your reasoning to discuss the results.

10. Fig.7: It seems that author just take screenshot and put in manuscript. Same comment on Fig.8, here we can see the top bar from software itself. Please provide figures in a better manner.

11. Provide a quantitative and qualitative comparison between passive and active cooling method for different cases.

Please eliminate grammatical errors and misleading sentences. 

Author Response

Many thanks for the review.

According to the reviewer’s suggestions, the authors have made appropriate changes in the article. They have also tried to answer the reviewer’s questions.

I am bit confused about the title of manuscript. Is it review article or research article? Authors presented some research results in manuscript after giving some literature review. Please have a better title.

The article has both the traits of a review article and a research article. The authors have highlighted a research gap in the issue of the influence of surface texture on heat dissipation.

An adequate literature review closely related to this issue was carried out, which is why the manuscript has “review” in the title. In addition, the authors were interested in
the topic and carried out analyses related to it. Furthermore, the issue has a developmental character. Therefore, the authors would like to keep the current title of the paper.

Table 1 and Table 2: provide the references for properties.

Appropriate references have been added

Fig.1(a) and Fig.2 are kind of same. Please have only one figure.

Appropriate changes have been made

Table 3: Any criteria to define geometrical dimensions of the heat sink? How did author define them please provide explanations.

The authors followed two criteria when selecting the dimensions of the heat sink;

The first criterion was the dimensional unification of the standard heat sink used in the electronics industry. The dimensions used are the actual dimensions obtained by reverse engineering.

The second criterion was the possibility of making a heat sink (as a test object) with these geometrical dimensions in such a way that it would be possible to shape the surface using cavity methods in a manner defined by the authors. The study of a heat sink made in this way is a separate paper and the results will be published in the near future.

In Fig. 3, show all the defined parameters in Table3.

Appropriate changes have been made.

Some Nomenclature are not provided in manuscript. It is difficult to know the meaning of parameters author used. Could author provide a Table for Nomenclatures?

We realize that in some journals articles are preceded by a table of designations, nomenclature. MDPI Publisher does not have such requirements, for this reason we do not present such a table. However, the nomenclature has been explained in the text in accordance with the reviewer's suggestion.

Line 423: The graph - Figure 6. It should be only Figure (6).

A correction has been made.

What is dotted line in Fig.6 and Fig. 10.

Appropriate changes have been made.

Author considered all the mode of heat transfer. Could you provide the contribution of each mode for heat transfer? It is important to support your reasoning to discuss the results.

We hope that the reviewer will accept the work without changes in this regard. Preparing an appropriate comparison at this stage requires repeating part of the research mainly concerned with simulations, which we cannot do in such a short time.

Fig.7: It seems that author just take screenshot and put in manuscript. Same comment on Fig.8, here we can see the top bar from software itself. Please provide figures in a better manner.

The authors agree with the reviewer. However, while making corections in the text, taking into account suggestions from other reviewers, the figures have been modified.

Provide a quantitative and qualitative comparison between passive and active cooling method for different cases.

According to the authors, comparisons of passive and active cooling are already included in the text. A quantitative comparison can be found in Chapter 4 -conclusions lines 528-533 and 540-546.

On the other hand, the qualitative comparison is not formulated explicitly, but the data presented in Figure 5 and Figure 8, in our opinion, in this aspect should be satisfactory.

Reviewer 2 Report

I think the optimization method for heat transfermation should be considered, such as surrogate assisted optimization, evolutionary optimization and deep learning methods.

Moreover, some important papers should be considered, such as:

High-dimensional model representation-based global sensitivity analysis and the design of a novel thermal management system for lithium-ion batteries.

A surrogate assisted thermal optimization framework for design of pin-fin heat sink for the platform inertial navigation system.

Thermal reliability-based design optimization using Kriging model of PCM based pin fin heat sink.

Image-based reconstruction for a 3D-PFHS heat transfer problem by ReConNN

It seems good.

Author Response

Many thanks for the review.
Apropriate changes in the article have been made in accordance to the reviewer’s suggestions.

Many thanks for Your suggestions aiming at improvement of the article’s quality.

The issue of heat transfer optimisation, particularly concerning surface shaping, will be a separate article and it is not planned to present the optimisation results in this work. The authors are finalising their research on this issue. However, the authors are grateful for drawing attention to this issue, as it is of scientific interest.

The papers suggested by the Reviewer have been cited accordingly. A suitable line or a sentence has been added in the text for each one of them. This new references contributed to the scientific quality of the article.

Reviewer 3 Report

This paper is appropriate for publication in this journal because the contents and conclusions are well organized. The following are my suggestions. Kindly consider.

-        In abstract (line 29) and in modeling (page 7), you mentioned radiation heat transfer of the heat sink. But in your simulation, you neglected this heat transfer phenomenon. Please consider and revise the manuscript.

-        Nothing new from figure 8, please omit it.

-        Because of a review paper, you must discuss work by work rather than a cluster of citation, e.g., “The type of cooling system used - passive [4-20] / active (forced) [20-48] and the selection of material [11-15,33-37,49-51],” etc. A tabular form with information in details is suggested to clarify state-of-the-art studies in heat sink.  

-        You stated, “The working space of the heat sink for passive cooling is an open space”, can I see a thermal plume of the airflow?

Author Response

Many thanks for the review
According to the reviewer’s suggestions, the authors have made appropriate changes in the article. They have also tried to answer the reviewer’s questions.

This paper is appropriate for publication in this journal because the contents and conclusions are well organized. The following are my suggestions. Kindly consider.

In abstract (line 29) and in modeling (page 7), you mentioned radiation heat transfer of the heat sink. But in your simulation, you neglected this heat transfer phenomenon. Please consider and revise the manuscript.

The authors are not sure they understood this reviewer's remark correctly. All three types of thermal energy transfer were taken into account during simulations. All the data presented in Chapter 3 take into account all types of heat transfer together. Some minor changes have also been made in the text regarding this issue.

Nothing new from figure 8, please omit it.

The Authors agree with the Reviewer. Figure 8 has been deleted.

Because of a review paper, you must discuss work by work rather than a cluster of citation, e.g., “The type of cooling system used - passive [4-20] / active (forced) [20-48] and the selection of material [11-15,33-37,49-51],” etc. A tabular form with information in details is suggested to clarify state-of-the-art studies in heat sink.

According to the authors, the citation groups allow the reader to sort the references concerning some of the aspects to which the cited articles relate. There are 10 citation groups in the paper. They relate to the following aspects addressed in the articles:

1) Active cooling,

2) Passive cooling,

3) Type of material used for the construction of the heat sink,

4) Effect of ambient conditions on the performance of the heat sink,

5) Influence of manufacturing method on the performance of the heat sink,

6) Whether the heat sink in question has been used in electronics (this is undoubtedly the most common application for small cooling systems)

7) Whether a physical experiment was conducted,

8) Whether simulation studies were conducted,

9) Was an FEM analysis performer,

10) Was the effect of the heat sink geometry on its performance discussed.

E.g. The article by Hoang et al. Hoang, C.H.; Fallahtafti, N.; Rangarajan, S.; Gharaibeh, A.; Hadad, Y.; Arvin, C.; Sikka, K.; Schiffres, S.N.; Sammakia, B. Impact of Fin Geometry and Surface Roughness on Performance of an Impingement Two-Phase Cooling Heat Sink. Applied Thermal Engineering 2021, 198, 117453, doi:10.1016/j.applthermaleng.2021.117453.) was referred to in citation groups 2), 4), 5), 7), 8), 9), 10) because the work dealt with all the aspects mentioned. In addition, a paragraph was dedicated to this articel. Lines 257-263, in which more details of the work are described.

This is how most of the references were treated.

The reviewer is right about the prefferable form of presentation of the citation group. The authors prepared such a table while writing the paper. However, the authors think that this table is rather too large and it can not be reasonably included in the paper. At the request of the reviewer, we present an excerpt of it.

(attachment)

You stated, “The working space of the heat sink for passive cooling is an open space”, can I see a thermal plume of the airflow?

At the reviewer's request, we present the thermal plume for open space in passive cooling.

(attachment)

Round 2

Reviewer 3 Report

Although, the authors attempted to revise the manuscript. The current version of the manuscript is needed extra revision. 

1. Check line 313. 

2. The inputted parameters in Table 1 is not enough to simulate radiation, e.g., emissivity missed. 

A model for radiation computation must be stated, e.g., DO, S2S, etc. 

3. BC for computational domains (both active cooling and passive cooling) must be displayed in figures. 

These are very important so that the reader can reproduce your work. 

Minor editing of English language required. 

Author Response

Response to a review

The authors would like to thank once again for drawing attention to important issues in the article.

  1. Check line 313.

The authors corrected the markings contained in the line of text pointed by the Reviewer.

  1. The inputted parameters in Table 1 is not enough to simulate radiation, e.g., emissivity missed. A model for radiation computation must be stated, e.g., DO, S2S, etc.
  2. BC for computational domains (both active cooling and passive cooling) must be displayed in figures. These are very important so that the reader can reproduce your work.

The authors fully agree with the Reviewer. All necessary information has been completed.

Appropriate changes have been made to Table 1. Radiation heat transfer model has been steated (lines 344-349).

Boundary conditions for both cases of cooling were supplemented, which are included in the modified Table 2.

Changes made according to the reviewer’s sugestions significantly inprooved the quality of the paper.
